# Dishevelled2 activates WGEF via its interaction with a unique internal peptide motif of the GEF
Aishwarya Omble[1,2], Shrutika Mahajan[1], Ashwini Bhoite[1,2] & Kiran Kulkarni [1,2] ✉

The Wnt-planar cell polarity (Wnt-PCP) pathway is crucial in establishing cell polarity during development and tissue homoeostasis. This pathway is found to be dysregulated in many pathological conditions, including cancer and autoimmune disorders. The central event in Wnt-PCP pathway is the activation of *Weak-similarity guanine nucleotide exchange factor* (WGEF) by the adapter protein Dishevelled (Dvl). The PDZ domain of Dishevelled2 ($Dvl2^{PDZ}$) binds and activates WGEF by releasing it from its autoinhibitory state. However, the actual $Dvl2^{PDZ}$ binding site of WGEF and the consequent activation mechanism of the GEF have remained elusive. Using biochemical and molecular dynamics studies, we show that a unique "internal-PDZ binding motif" (IPM) of WGEF mediates the WGEF-$Dvl2^{PDZ}$ interaction to activate the GEF. The residues at $P_2, P_0, P_{-2}$ and $P_{-3}$ positions of IPM play an important role in stabilizing the $WGEF^{pep}$-$Dvl2^{PDZ}$ interaction. Furthermore, MD simulations of modelled $Dvl2^{PDZ}$-$WGEF^{IPM\ peptide}$ complexes suggest that WGEF-$Dvl2^{PDZ}$ interaction may differ from the reported $Dvl2^{PDZ}$-IPM interactions. Additionally, the *apo* structure of human $Dvl2^{PDZ}$ shows conformational dynamics different from its IPM peptide bound state, suggesting an induced fit mechanism for the $Dvl2^{PDZ}$-peptide interaction. The current study provides a model for Dvl2 induced activation of WGEF.

Wnt signalling pathways regulate diverse cellular functions during embryonic development and tissue homoeostasis[1]. The central event in Wnt pathways involves activation of hepta helical membrane receptors, Frizzleds, by the secreted glycoproteins, Wnts. The signalling further propagates on the subsequent recruitment of a scaffold protein, Dishevelled (Dvl), by the C-terminal target binding domain of the Frizzled receptors. This signal further diverges into two modes; canonical β-catenin-dependent signalling and non-canonical β-catenin-independent signalling. The canonical β-catenin-dependent signalling promotes the expression of target genes by facilitating the nuclear translocation of β-catenin and subsequent activation of TCF/LEF transcription factors associated with those target genes. This mode of Wnt signalling is majorly involved in cellular processes like cell proliferation[2]. On the other hand, the non-canonical β-catenin-independent signalling is further subdivided into Wnt/Calcium and Wnt-planar cell polarity (Wnt-PCP) pathways[3,4]. These non-canonical pathways are shown to control cell polarity and migration[2]. Aberrations in the Wnt signalling pathways are implicated in many pathological conditions, including cancer, developmental and autoimmune disorders[5,6].

One of the key features of Wnt-PCP pathway is the regulation of actin cytoskeleton reorganization through the activation of Rho GTPases, namely RhoA, Rac and Cdc42[7]. Rho GTPases function as molecular switches that cycle between GTP bound ON (active) and the GDP bound OFF (inactive) states[8]. This switching between ON and OFF states is brought about by the regulatory proteins known as Guanine nucleotide exchange factors (GEFs) and GTPase activating proteins (GAPs), respectively[9]. Appropriate signalling cues, transducing from the membrane receptors such as G-protein coupled receptors (GPCRs)[10], receptor tyrosine kinase[11] and integrin receptors[12], activate their downstream GEFs, which in turn trigger Rho GTPases by exchanging their bound GDP with GTP. However, the majority of GEFs exist in an autoinhibitory state and require other signalling proteins for their activation[10]. In the context of Wnt-PCP pathway WGEF (Weakly similar to RhoGEF 5/TIM), also known as Arhgef19/ Ephexin2), activates RhoA[13,14]. In *Xenopus*, WGEF-RhoA exerts convergent extension during gastrulation. Thus, marking the Wnt-PCP/RhoA pathway as one of the critical pathways involved in developmental and tissue regeneration processes in the higher eukaryotes[15–18]. Similar to canonical DH-PH GEFs, like TIM and Ephexin, WGEF consists of the conserved catalytic Dbl homology

[1]Division of Biochemical Sciences, CSIR-National Chemical Laboratory, Dr. Homi Bhabha Road, Pune 411008, India. [2]Academy of Scientific and Innovative Research (AcSIR), Ghaziabad 201002, India. ✉e-mail: ka.kulkarni@ncl.res.in

(DH) domain responsible for its GEF activity and the Pleckstrin homology (PH) domain, which is involved in the membrane localization of the GEF[13]. Furthermore, WGEF consists of a C-terminal Src homology 3 (SH3)[13] and an N-terminal inhibitory (NID) domain. It has been shown that the phosphorylation of a conserved tyrosine in the NID region is critical in the activation of the GEF[19–21]. Besides NID, the SH3 domain also contributes to the autoinhibition of the GEF through its intra- and intermolecular interactions[21–23].

In the context of the Wnt-PCP pathway, the adapter protein Dishevelled is shown to be involved in releasing the autoinhibitory state of WGEF[17]. Dishevelled (Dvl), which has three paralogs in humans, viz., Dvl1, Dvl2 and Dvl3, operate as a signalling hub in exerting both canonical and non-canonical Wnt signalling pathways[24,25]. This protein consists of three conserved domains, namely, an N-terminal DIX (Dishevelled-Axin) domain, a central PDZ (PSD-95, DLG, ZO1) domain and a C-terminal DEP (Dishevelled, EGL-10, Pleckstrin) domain[26]. These domains play diverse roles during the Wnt signalling pathways[27–30]. Regarding the activation of WGEF, the PDZ domain of Dishevelled2 (Dvl2) is shown to be the key player. Dvl2 binds to WGEF through its PDZ domain to release the GEF from its autoinhibitory state[17]. However, the exact region of WGEF that interacts with Dvl2$^{PDZ}$ and its mode of interaction with Dvl2 have remained elusive. In the present study, we have identified an evolutionary conserved Dvl2$^{PDZ}$ binding motif in WGEF (hereafter referred to as WGEF$^{pep}$), which facilitates the activation of the GEF by disrupting its autoinhibitory state. Using biochemical assays and molecular dynamics simulation studies, we have elucidated the mode of Dvl2$^{PDZ}$–WGEF interaction and its consequence on the activation of the GEF.

## Results

### A novel N-terminal conserved 'internal peptide' motif of WGEF mediates its interaction with Dvl2$^{PDZ}$

Previously, Igor et al. have shown that the PDZ domain of Dvl2 binds to the N-terminal domain of WGEF, resulting in activation of the GEF. Further, deletion of the first 213 residues from the N-terminal of human WGEF (hWGEF$^{\Delta213}$) has been shown to reduce its affinity for Dvl2 considerably but at the same time to significantly enhance its GEF activity. Thus, it was suggested that the N$^{213}$ region of the GEF could be involved in WGEF–Dvl2$^{PDZ}$ interaction[17]. In another study, hWGEF$^{\Delta302N-term}$ mutant is shown to be free from autoinhibition, suggesting that its PDZ binding motif precedes the autoinhibitory domain of the GEF (spanning residues 291-NSVLYQEY-298)[17,21]. However, the exact motif of WGEF that binds to Dvl2$^{PDZ}$ is unclear from these studies. Like other PDZ domains, the Dvl2$^{PDZ}$ also exhibits promiscuity in terms of the peptide motifs that it binds to. The peptide motifs that PDZ domains recognize are primarily *C-terminal peptide motifs*, which are composed of a stretch of 7–10 residues situated at the C-terminus of the binding partners. Apart from these, some PDZ domains can also recognize the *internal motifs* that possess a similar number of residues but can lie anywhere in their binding partners[31–33]. Binding of PDZ with internal motifs mimic that of the C-terminal peptide binding with slight variations in the residues of the peptide involved in the binding[33,34].

Although not very stringent, there is some consensus in the sequences of *C-terminal peptide motifs* that bind to PDZ domains[31,32]. However, there is no statistically significant data to arrive at a consensus sequence for the PDZ domain binding *internal peptide motifs*. Therefore, to identify the Dvl2$^{PDZ}$ interacting *internal motif* of WGEF, we searched the sequences of this family of GEFs with the putative Dvl2$^{PDZ}$ binding motifs identified from peptide-phage display studies[35]. We employed three prominent peptide motifs, X-Y-G-W-Φ$^a$-D/G, X-W-Φ$^a$-D-G-P and W-Φ$^s$-D-X-P (where X, Φ$^a$ and Φ$^s$ are any, aliphatic and short side chain hydrophilic amino acids [S/T], respectively), as templates for the search. The only hit, 349-GSTFSLWQDIP-359 (hereafter referred to as WGEF$^{pep}$), obtained from the sequence search, lies between the DH and the inhibitory domain of hWGEF (Fig. 1a). This particular motif is highly conserved amongst WGEFs family, indicating it to be the most putative Dvl2$^{PDZ}$ binding site present in WGEFs (Fig. 1a and

Supplementary Fig. 1). This observation is contrary to the previous study in which the Dvl2$^{PDZ}$ binding was mapped towards the N-terminus of hWGEF (between amino acid residues 1–213)[17], whereas the motif identified by us is present between amino acid residues 349–359 of hWGEF. Hence, to validate our observations, we designed constructs of the GEF comprising both the autoinhibitory and the Dvl2$^{PDZ}$ binding motif and yet excluding the major portion of its N-terminus. Since some of these hWGEF constructs were found to be insoluble and/or poorly expressed in *E. coli*, we used WGEF from *Xenopus laevis* (xWGEF) as a surrogate system for further studies. To test the influence of Dvl2$^{PDZ}$ binding on the catalytic activity of WGEFs, we performed nucleotide exchange assays on these constructs, both in the presence and absence of the PDZ domain. Clearly, the xWGEF330 (equivalent to hWGEF272) construct, which is composed of both the inhibitory and the PDZ binding domains, showed insignificant GEF activity. However, in the presence of 10-fold excess (50 μM) of Dvl2$^{PDZ}$, there is a fourfold increase in its GEF activity (Fig. 1b, c). It is worth noting that the lower concentration of Dvl2$^{PDZ}$ (25 μM) did not show any observable change in the GEF activity. These observations suggest that the binding of Dvl2$^{PDZ}$ indeed promotes the activity of GEF; however, the affinity of the xWGEF330 for Dvl2$^{PDZ}$ is considerably low (Fig. 1b, c). Next, to show that the identified internal peptide motif of WGEF, 402(349)-GSTFSLWQDIP-412(359)/ WGEF$^{pep}$ [residue numbers shown in the parentheses corresponds to hWGEF], binds to Dvl2$^{PDZ}$, we performed the GEF assays in the presence of both Dvl2$^{PDZ}$ and WGEF$^{pep}$. Since the internal peptides are shown to have relatively low affinity for PDZ domains, we used a fourfold molar excess of WGEF$^{pep}$ (200 μM) to disrupt the WGEF–Dvl2$^{PDZ}$ interaction (Fig. 1b, c). From this exercise, it is clear that WGEF$^{pep}$ competitively binds to the PDZ domain and replaces the WGEF in Dvl2$^{PDZ}$–WGEF interaction, causing a decrease in the catalytic activity of WGEF. Thus, our observations unambiguously suggest that the 402–412 region of xWGEF is the 'internal peptide motif' that mediates its interactions with Dvl2$^{PDZ}$.

### WGEF$^{pep}$ motif–Dvl2$^{PDZ}$ interaction is sufficient to partially release the autoinhibition of WGEF

Canonically, PDZ domains bind to *C-terminal* peptides, and therefore, the residue at the extreme C-terminal of the peptide is referred to as the 0th or $P_0$ residue. The remaining residues towards the N-terminus of the peptide are numbered in reverse order as '−1', '−2', '−3' or $P_{−1}$, $P_{−2}$, $P_{−3}$, and so on[36–38]. Dvl2$^{PDZ}$ binds to both *C-terminal* as well as the non-canonical *internal peptides*[35,39]. Therefore, we will refer to the residue that interacts with the X-Φ-G-Φ motif of the carboxylate binding loop (X represents any amino acid and Φ represents hydrophobic residues)[31] as $P_0$, followed by $P_{−1}$, $P_{−2}$, $P_{−3}$ for residues preceding $P_0$ and $P_1$, $P_2$, $P_3$ for the residues that lie towards the C-terminus.

Zhang et al. have reported earlier that the side chain of aspartate residue present in the internal peptide mimics the free C-terminal carboxylate group and binds to the carboxylate binding loop of Dvl2$^{PDZ}$. Thus, this particular interaction is asserted to stabilize the Dvl2$^{PDZ}$-effector binding[35]. Therefore, to test whether the WGEF$^{pep}$ follows this "usual" internal peptide–Dvl2$^{PDZ}$ interaction, we substituted the aspartate present at the $P_0$$^{th}$ (Fig. 1d) position of the peptide with alanine (WGEF$^{pepD9A}$). Interestingly, with an excess amount of WGEF$^{pepD9A}$ peptide, there was a decrease in the exchange activity of WGEF, but not to an extent brought out by the wild-type WGEF$^{pep}$ (Fig. 1b, c). We performed a comparable exercise on GEF by replacing this particular aspartate (xWGEF330$^{D410A}$) in xWGEF construct. Since this xWGEF mutant was found to be insoluble, no further studies could be conducted. Other studies also suggest that proline, tryptophan and leucine at $P_2$, $P_{−2}$ and $P_{−3}$ positions, respectively, also play a key role in stabilizing Dvl2$^{PDZ}$–effectors interactions[35]. To test the role of these residues when they are part of the GEF, we produced xWGEF330$^{P412A}$, xWGEF330$^{W408A}$ and xWGEF330$^{L407A}$ mutants; however, only the xWGEF330$^{L407A}$ was found to be soluble. Surprisingly, the enhanced GEF activity of xWGEF330$^{L407A}$ is independent of its interaction with Dvl2$^{PDZ}$ (50 μM) (Fig. 1b, c), as the addition of Dvl2$^{PDZ}$ (50 μM) does not

**Fig. 1 | Activity studies of xWGEF330 and xWGEF330 mutants. a** Schematic representation of WGEF domains (N-terminal Inhibitory: NID, Dbl homology: DH, pleckstrin-homology: PH and src Homology-3: SH3) and sequence alignment of the conserved PDZ binding motif of Zebrafish (Uniprot ID: E7EY470), Xenopus (Uniprot ID: A5X5J0), Mouse (Uniprot ID: Q8BWA8), Human (Uniprot ID: Q8IW93-1) and Bovine (Uniprot ID: E1BQ24) ahead of DH domain in WGEF. The amino acid residue numbers in the upper and lower parts of the diagram correspond to hWGEF and xWGEF, respectively. **b** GEF assay plots of xWGEF330, xWGEF330$^{L407A}$ and xWGEF330$^{Y353E}$, in the presence and absence of Dvl2$^{PDZ}$ and peptide variants **c** Bar graph showing relative GEF activities of xWGEF variants normalized against that of wild-type (autoinhibited xWGEF330) protein. The error bar and * represent standard deviation (±SD) and significance as ****$P < 0.0001$, ***$P < 0.001$, **$P < 0.01$ and *$P \leq 0.01$, respectively. **d** Sequence alignment of Dvl2$^{PDZ}$ binding internal peptides and WGEF internal peptide sequence. Residue positions are labelled in red.

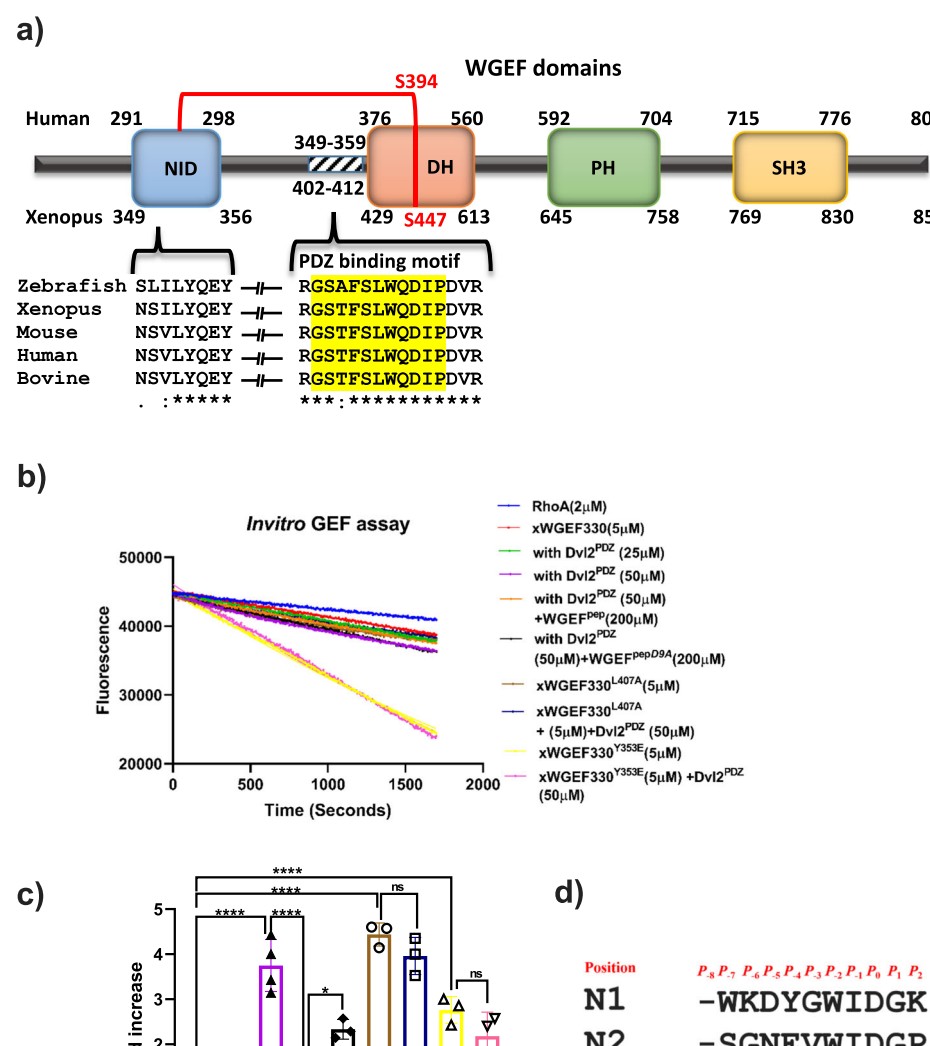

seem to affect its GEF activity. The higher GEF activity observed for xWGEF330$^{L407A}$ could be due to the structural rearrangement caused by the mutation. However, from our experiments it is not clear how L407A substitution influences the autoinhibition of the GEF. Thus, this aspect requires further investigations.

Next, to investigate whether the WGEF–Dvl2$^{PDZ}$ interaction promotes the GEF activity by releasing its autoinhibition or it follows some unknown activation mechanism, we generated a Y353E mutant of xWGEF330. This particular tyrosine (Y$^{353xWGEF/295hWGEF}$) of hWGEF is present in NID and locks the GEF by interacting with its DH domain. Furthermore, hWGEF$^{Y295E}$ is shown to exist in autoinhibition-free form[21]. Interestingly, the xWGEF330$^{Y353E}$ mutant exhibited a nearly threefold increase in the GEF activity compared to its wild-type form. Further, the presence of Dvl2$^{PDZ}$ (50 μM) did not alter the GEF activity of the mutant (Fig. 1b, c). Thus, these results suggest that the Dvl2$^{PDZ}$ binds to the WGEF$^{pep}$ region of the GEF, lying between its inhibitory (NID) and the DH domains. Furthermore, this particular interaction between WGEF and Dvl2$^{PDZ}$ activates the GEF by partially releasing it from its autoinhibitory state regulated by the NID. The

mechanism through which the autoinhibition governed by the SH3 domain is released remains unknown.

## Binding of WGEF$^{pep}$ with Dvl2$^{PDZ}$ is mediated by the residues at the $P_{-2}$, $P_0$ and $P_2$ positions of the binding motif

Although the WGEF$^{pep}$ region is highly conserved in the WGEF family (Supplementary Fig. 1), the interaction between WGEF and Dvl2$^{PDZ}$ appears to be feeble. This aligns with the observations made for other PDZ–internal peptide interactions[33,35]. Perhaps differential affinity of the Dvl with its effectors is one of the strategies adopted by Wnt-Frizzled pathways to exert diverse signalling modes[25,30]. Although our GEF assay-based studies have shown that, indeed, the residue at the $P_0$ position of WGEF$^{pep}$ plays a crucial role in mediating WGEF–Dvl2$^{PDZ}$ binding, it is not clear how variations in other residue positions influence the interaction. Therefore, to obtain the structural basis of Dvl2$^{PDZ}$–WGEF interaction, initially, we tried to crystallize this protein complex. Despite extensive efforts, crystallization trials met with no success. Hence, we resorted to crystallizing the WGEF$^{pep}$ (peptide) in complex with Dvl2$^{PDZ}$. However, even this attempt did not yield

**Table 1 | Data collection and refinement statistics for the structures of human Dishevelled2 PDZ domain, co-crystallized and fused with WGEF$^{pep}$, respectively**

| | hDvl2$^{PDZ}$ crystallized with WGEF$^{pep}$ | hDvl2$^{PDZ}$ fused with WGEF$^{pep}$ |
|---|---|---|
| **Data collection** | | |
| Source | ID23-2 beamline of ESRF | BL21-PX beamline of Indus-2 synchrotron |
| Space group | P4$_3$2$_1$2 | I 4$_1$ |
| a (Å) | 45.98 | 59.49 |
| b (Å) | 45.98 | 59.49 |
| c (Å) | 78.42 | 58.25 |
| α (°) | 90 | 90 |
| β (°) | 90 | 90 |
| γ (°) | 90 | 90 |
| Resolution limits (Å) | 32.51–1.75 | 42.07–3.00 |
| R$_{merge}$ | 0.067(0.731) | 0.058(1.091) |
| I/s (I) | 23.3(3.8) | 27.8(2.7) |
| Number of reflections | 157,526(7686) | 30,769(5090) |
| Unique reflections | 9023(482) | 2071(336) |
| Completeness (%) | 100(100) | 99.9(100) |
| Multiplicity | 17.5(15.9) | 14.9(15.1) |
| CC(1/2) | 0.999(0.915) | 1(0.731) |
| **Refinement** | | |
| Resolution limits (Å) | 32.51–1.75 | 29.75–3.00 |
| Number of reflections | 8984 | 2067 |
| Working set | 8594 | 1951 |
| Test set | 390 | 116 |
| R$_{work}$/R$_{free}$ | 0.227/0.252 | 0.299/0.335 |
| **Number of atoms** | | |
| Protein | 1245 | 512 |
| Water | 20 | |
| **B factors** | | |
| Protein atoms (Å$^2$) | 39.26 | 114.32 |
| Water | 37.25 | |
| **RMSD from ideal values** | | |
| Bond length (Å) | 0.007 | 0.009 |
| Bond angles (°) | 1.089 | 1.361 |
| **Ramachandran plot** | | |
| Preferred (%) | 97.53 | 87.32 |
| Allowed (%) | 2.47 | 12.68 |
| PDB code | 8WWR | 8YR7 |

Single crystal was used for data collection. Values in parenthesis are for the highest-resolution shell.

the desired result, as there was no electron density for the bound peptide in the crystal structure of the WGEF$^{pep}$–Dvl2$^{PDZ}$ complex (PDB ID: 8WWR) (Table 1). Furthermore, we tried to obtain the WGEF$^{pep}$–Dvl2$^{PDZ}$ complex structure by fusing the peptide with the C-terminus of the PDZ domain. Even this approach failed to produce the WGEF$^{pep}$–Dvl2$^{PDZ}$ complex structure (PDB ID: 8YR7) (Supplementary Fig. 2 and Table 1). Therefore, we systematically designed divergent peptides of WGEF$^{pep}$ and biochemically probed their interaction with Dvl2$^{PDZ}$. The closest homologue of WGEF$^{pep}$ is an engineered peptide, N3$^{pep}$, whose structure in complex with Dvl2$^{PDZ}$ is available (PDB ID: 3CC0). Hence, by using this structure as template, we designed WGEF$^{pep}$ variants by successively substituting its residues from $P_{-3}$ to $P_2$ positions with alanine (Fig. 1d), as these residues are

seen to be lying within 5 Å from Dvl2$^{PDZ}$ residues and they are conserved between WGEF$^{pep}$ and N3$^{pep}$. To assess the differential contributions from these residues of the peptide, we determined the dissociation constants of the WGEF$^{pep}$ variants for Dvl2$^{PDZ}$, employing Microscale Thermophoresis (MST). In addition to the WGEF$^{pep}$ variant peptides the minimal hWGEF (330–581) construct, obtained from our solubility screening, was used for the MST study (Fig. 2a, b, d and Supplementary Fig. 3). It is interesting to note that the affinities of WGEF$^{pep}$ and N3$^{pep}$ are nearly identical for Dvl2$^{PDZ}$, with dissociation constants ($K_d$) of $45 \pm 5$ and $54 \pm 7$ μM, respectively (Fig. 2a, d and Table 2). This further validates the choice of our model for delineating WGEF–Dvl2$^{PDZ}$ interaction. Substitution of Gln at the $P_{-1}$ (Fig. 1d) with Ala (WGEF$^{pepQ8A}$) enhances the affinity of the peptide by almost 6-fold, whereas Ile at $P_1$ to Ala (WGEF$^{pepI10A}$) does not seem to affect the WGEF$^{pep}$–Dvl2$^{PDZ}$ interaction (Fig. 2b, d and Table 2). It is interesting to note that amongst these two residues, only the side chain of Ser at $P_{-1}$ makes direct interaction with Dvl2$^{PDZ}$ in the Dvl2$^{PDZ}$–N3$^{pep}$ complex crystal structure. On the contrary, the substitution of Leu at the $P_{-3}$ position (WGEF$^{pepL6A}$) significantly reduces the affinity of the peptide for Dvl2$^{PDZ}$ (Fig. 2c, d and Table 2). Furthermore, the substitution of tryptophan (WGEF$^{pepW7A}$), aspartate (WGEF$^{pepD9A}$) and proline (WGEF$^{pepP11A}$) residues at the $P_{-2}$, $P_0$ and $P_2$ positions, respectively, with alanine completely abolishes the WGEF–Dvl2$^{PDZ}$ interactions (Table 2 and Supplementary Table 1). Thus, MST studies suggest that the residues at $P_{-2}$, $P_0$ and $P_2$ positions (W, D & P) are crucial for the WGEF-Dvl2$^{PDZ}$ binding, whereas leucine at the $P_{-3}$ position may influence the peptide–Dvl2$^{PDZ}$ interaction either by stabilizing the complex or by enhancing the solubility of the peptide.

## Internal peptide ligands of Dvl2$^{PDZ}$ exhibit divergent modes of interaction

From GEF activity and MST studies, we have identified the residues of the *internal peptide motif* that drive the WGEF–Dvl2$^{PDZ}$ interaction. Furthermore, in both canonical and non-canonical (internal) peptides, the ligand adopts antiparallel β strand conformation and forms a β sheet by associating with the β2 strand of the PDZ domain[40]. Additionally, in the canonical binding mode, the C-terminus of the $P_0$ residue engages with the X-Φ-G-Φ motif of the carboxylate binding loop to stabilize the complex (Fig. 1a, d). However, in the non-canonical PDZ domain binding, the *carboxylate binding loop* may not adopt a closed conformation and instead engages with the internal peptide through a network of hydrogen bonds involving either the main chain or the side chain atoms of the peptide[35,37,41]. Due to the lack of structural information on the WGEF$^{pep}$–Dvl2$^{PDZ}$ complex, their binding mode has remained unclear. Hence, to address this aspect, we performed a total of 9 μs molecular dynamics simulations of the Dvl2$^{PDZ}$ structure, complexed with WGEF$^{pep}$, WGEF$^{pepQ8A}$ and N3$^{pep}$ peptides. Since for two of the former peptides, there was no crystal structure available, we modelled the respective Dvl2$^{PDZ}$–peptide complexes using the coordinates of the *apo* (PDB ID: 8WWR) (Table 1) and Dvl2$^{PDZ}$-N3$^{pep}$ structures.

To understand the role of every residue in stabilizing the interaction, we plotted residue-wise peptide–PDZ domain interaction propensity (probability of the residue side-chain lying within 4 Å from the PDZ residue) (Fig. 3a–d). Propensity and the residue RMSF (Supplementary Fig. 4) plots obtained from MD simulations show that residues at positions $P_{-2}$, $P_{-1}$, $P_0$, $P_1$ and $P_2$ play significant roles in stabilizing the peptide–Dvl2$^{PDZ}$ interaction. Amongst these, the aspartate residue of the $P_0$ position of both WGEF$^{pep}$ and N3$^{pep}$ forms maximum number of interactions with X-Φ-G-Φ motif of the *carboxylate binding loop* (Fig. 3a, b, e, f). Apart from Asp at the $P_0$ position, it was observed that residues at the positions $P_{-1}$, $P_{-2}$ and $P_{-3}$ also make interactions with the residues belonging to the β2-α2 region of Dvl2$^{PDZ}$ (Fig. 3a, b, e, f). Interestingly, most interactions between WGEF$^{pep}$-Dvl2$^{PDZ}$ are similar to those observed in the Dvl2$^{PDZ}$-N3$^{pep}$ (Fig. 3a, b). However, subtle differences in the WGEF$^{pep}$ residues at $P_{-4}$ and $P_{-5}$ exhibit divergence in their interaction with the PDZ domain, when compared with those of N3$^{pep}$ interactions. In Dvl2$^{PDZ}$–N3$^{pep}$ complex residues at $P_{-4}$ and $P_{-5}$ positions interact with β2 strand of Dvl2$^{PDZ}$ domain, which is absent in WGEF$^{pep}$-Dvl2$^{PDZ}$ simulated model (Fig. 3a, b, e, f). Perhaps that is why in

**Fig. 2 | Binding studies of Dvl2$^{\text{PDZ}}$ with different internal peptides and human WGEF. a** Binding curve corresponding to the interaction of Dvl2$^{\text{PDZ}}$ with WGEF$^{\text{pep}}$ and N3$^{\text{pep}}$. **b** and **c** Binding curve corresponding to WGEF$^{\text{pepQ8A}}$, WGEF$^{\text{pepI10A}}$ peptides, hWGEF$^{330\text{-}581}$, and WGEF$^{\text{pepL6A}}$ peptide, respectively. **d** Bar plots showing dissociation constants between Dvl2$^{\text{PDZ}}$ and different peptides. The error bars represent $K_{\text{d}}$ confidence obtained from triplicates.

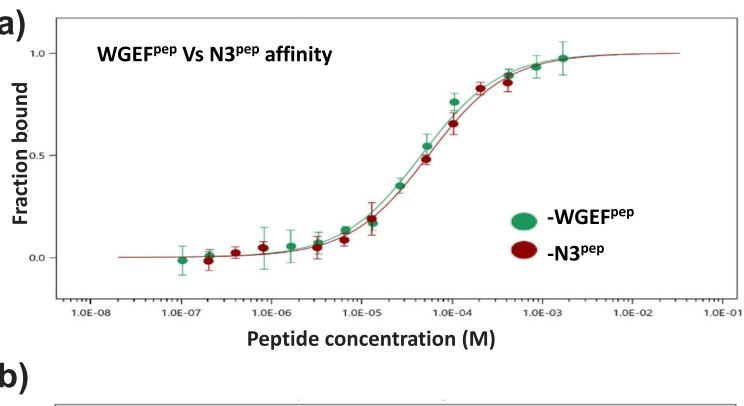

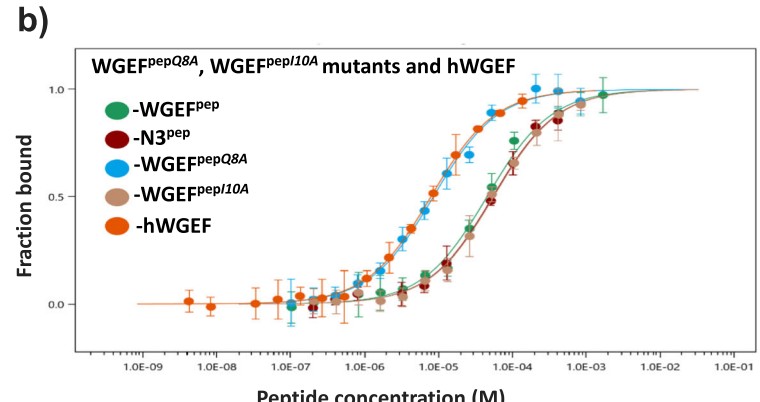

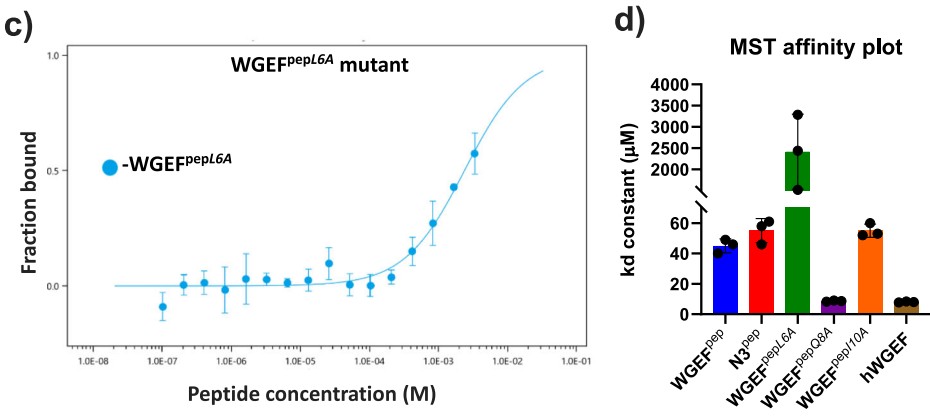

WGEF$^{\text{pep}}$-Dvl2$^{\text{PDZ}}$ simulations, residues from $P_{-4}$ onwards (towards N-terminus) exhibit higher RMSF values (Supplementary Fig. 4). Furthermore, the tryptophan at the $P_{-2}$ position of N3$^{\text{pep}}$ provides additional stacking interactions to stabilize the PDZ–peptide complex.

Our MST studies show that the substitution of Gln at $P_{-1}$ with Ala (WGEF$^{\text{pepQ8A}}$) in WGEF$^{\text{pep}}$ considerably increases the affinity of the mutant peptide towards Dvl2$^{\text{PDZ}}$ by 6-folds (Fig. 2b, d). Interestingly, the MD simulations on the WGEF$^{\text{pepQ8A}}$–Dvl2$^{\text{PDZ}}$ complex indicate a higher propensity for the Trp at the $P_{-2}$ position to stay in the hydrophobic groove formed by Ile 280, Leu 278, Leu 337, and Val 341 residues at the β2-α2 region of the PDZ domain (Fig. 3c, d, g). However, the backbone conformation of this residue that favours the hydrophobic interaction with the PDZ domain is governed by the residue at the $P_{-1}$ position. The presence of a residue with a longer side chain at $P_{-1}$ could restrict the conformational space of Trp at $P_{-2}$ and hence, may hinder the additional hydrophobic interactions between this residue and the PDZ domain (Fig. 3h). Thus, substitution of Gln with a shorter-side chain amino acid, like Ala, at $P_{-1}$ position promotes WGEF$^{\text{pep}}$–Dvl2$^{\text{PDZ}}$ interaction, as seen from MST studies. Thus, our MD studies provide insights into the Dvl2$^{\text{PDZ}}$–WGEF interaction and corroborate our biochemical studies on the Dvl2$^{\text{PDZ}}$–peptide complexes.

Furthermore, our MD studies and previously reported crystal structures of Dvl2-internal peptide complexes show no conserved mode of interaction between the internal peptides and Dvl2$^{\text{PDZ}}$. In the Dvl2$^{\text{PDZ}}$-N3$^{\text{pep}}$ crystal structure, it is observed that N3$^{\text{pep}}$ forms antiparallel β strand interaction with the β2 strand of the Dvl2$^{\text{PDZ}}$ domain (Supplementary Fig. 5a). Thus, to assess whether there is conservation of β sheet formation of the internal peptides with the β2 strand of the Dvl2$^{\text{PDZ}}$ domain, we calculated the secondary structural propensity of the peptide residues using their respective MD trajectories. In case of Dvl2$^{\text{PDZ}}$ domain and internal peptide complexes (WGEF$^{\text{pep}}$, WGEF$^{\text{pepQ8A}}$ and N3$^{\text{pep}}$ peptides), it was observed that these ligands did not show consistency in forming β sheet with the β2 strand of the Dvl2$^{\text{PDZ}}$ domain, as the ligand may not essentially adopt β strand conformation. Perhaps this loss of secondary structural conformation might aid the flexibility of the peptide and hence, enhance its interaction with Dvl2$^{\text{PDZ}}$ (Supplementary Fig. 5b–d).

## Dvl2$^{\text{PDZ}}$ exhibits ligand-dependent conformational changes

Earlier studies have shown that certain PDZ domains like Erbin$^{\text{PDZ}}$ and Syntrophin$^{\text{PDZ}}$ exhibit limited flexibility during peptide interaction. Perhaps that is why Erbin$^{\text{PDZ}}$ specifically accommodates C-terminal peptides with higher sequence specificity. However, Synrophin$^{\text{PDZ}}$ can interact with both

**Table 2 | Consolidated table showing dissociation constant ($K_d$) values for binding studies between different peptides, hWGEF protein and Dvl2$^{PDZ}$**

| Peptides | Sequence | $K_d$ (µM) |
|---|---|---|
| WGEF$^{pep}$ | GSTFSLWQDIP | 45 ± 5 |
| N3$^{pep}$ | EIVLWSDIP | 54 ± 7 |
| WGEF$^{pepL6A}$ | GSTFSAWQDIP | 2400 ± 900 |
| WGEF$^{pepW7A}$ | GSTFSLAQDIP | ND |
| WGEF$^{pepQ8A}$ | GSTFSLWADIP | 8.5 ± 0.8 |
| WGEF$^{pepD9A}$ | GSTFSLWQAIP | ND |
| WGEF$^{pepI10A}$ | GSTFSLWQDAP | 56 ± 4.3 |
| WGEF$^{pepP11A}$ | GSTFSLWQDIA | ND |
| hWGEF | Residues from 330 to 581 | 7.9 ± 0.3 |

the *C-terminal* as well as the *internal peptides*, provided the peptide undergoes a conformational change to form a β finger, in order to be incorporated within the peptide binding loop, as seen in the structure of nNOS–Syntrophin$^{PDZ}$ complex[42,43]. Another PDZ domain-containing protein, Par-6, displays allostery-mediated binding for C-terminal peptides, which exhibits an increase in affinity for the ligand by 10-folds upon binding of Cdc42 to its adjacent *Cdc42/Rac interactive binding* (CRIB) domain[44]. Par-6$^{PDZ}$ domain binds to the internal peptide of Pals1, as well, through a conformational change in the PDZ domain, which is independent of allosteric alterations[37]. Thus, it is clear from previous studies that PDZ domains exhibit huge conformational dynamics to accommodate their binding motifs. Earlier efforts have tried to characterize these dynamics using MD simulations[42]. It is worth mentioning that the dynamical features of PDZ domains presented from different MD studies extensively vary[45]. This could be due to variations in the type of PDZ structures (*apo*, or bound with C-terminal and/or internal peptide) employed in the MD simulations and the simulation time[45,46]. Since we have elucidated the *apo* structure of human Dvl2$^{PDZ}$ (PDB ID: 8WWR), we looked into the dynamics in its *apo* and different peptide-bound forms. It is important to mention that our hDvl2$^{PDZ}$ *apo* structure differs from the reported *Xenopus* Dvl2$^{PDZ}$ *apo* structures (PDB IDs: 3FY5 and 2F0A), with a substitution of just one amino acid (Human M$^{323}$/*Xenopus* I$^{310}$).

A comparison of available Dvl2$^{PDZ}$ structures shows deviations with RMSD values in the range of 1.7–2 Å. Importantly, significant conformational variations are observed in the α2 helix and β2–β3 loop, which are part of the peptide binding pocket. This conformational variation is more prominent in the structures of human Dvl2$^{PDZ}$, bound with different peptides (Fig. 4a–c and Supplementary Fig. 5a). This opens up the question of whether the peptide binding pocket of Dvl2$^{PDZ}$ is preformed or it exhibits peptide-induced conformational fit. To identify the amino acids that are dynamically coupled, we performed spectral decomposition of the *dynamic cross-coupling matrix (cij)* (Fig. 4d) obtained from all the MD simulations. As explained in the "Methods" section, the statistical coupling analysis (SCA) provides two main clusters of dynamically coupled amino acids, referred to as Sector 1 and Sector 2. These sectors can be further resolved into five *independent components* (ICs) (Supplementary Fig. 6). It is interesting to note that Sector 1 corresponding to all the simulations, which primarily consists of residues belonging to the peptide-binding pocket, show significant variations in terms of their residue composition and location of residues in the protein structure (Fig. 4e–h). Interestingly, Sector 1 of *apo* Dvl2$^{PDZ}$ is composed of residues residing in β2 and α2 regions of the domain, whereas Sector 1 of Dvl2$^{PDZ}$ internal peptide (WGEF$^{pep}$, WGEF$^{pepQ8A}$ and N3$^{pep}$) complexes had residues from the β2 strand, β1–β2 and β2–β3 loops (Fig. 4e–h). These findings further validate the internal allostery of PDZ domains as elucidated by other studies[47]. Additionally, variations in Sector 1 composition highlight peptide-dependent conformational adaptation of Dvl2$^{PDZ}$ domain. In contrast to the earlier study by Munz et al., we observe

that the α2 helix of the *apo* Dvl2$^{PDZ}$ structure has limited conformational flexibility rather than having the ability to explore all possible conformational space, specific to the ligand-bound forms. Perhaps, divergent observations of our studies from those published before could be due to the absence of a "true" *apo* structure of Dvl2$^{PDZ}$. Furthermore, the shorter duration of earlier MD simulation (200 ns)[42] might have exaggerated the conformational sampling of the PDZ domain. Thus, our structural and MD simulation studies suggest that Dvl2$^{PDZ}$ adopts peptide-dependent conformational change due to its intrinsic flexibility. In short, we can infer that Dvl2$^{PDZ}$ may not have a preformed binding pocket, rather it modulates the binding pocket based on the substrate available for binding. This adaptation is perhaps responsible for the peptide promiscuity of Dvl2$^{PDZ}$.

## Discussion

Despite nearly three decades of studies on PDZ domains, their modes of interaction with their binding partners have remained elusive. It is quite intriguing to learn that despite having highly conserved domain structure, PDZ domains exhibit divergent modes of interaction with their binding partners[31,42,48]. Dishevelled, being a signalling hub, recognizes diverse binding partners through its different domains, especially involving its PDZ domain[49]. Therefore, as seen in other PDZ domains, Dvl2$^{PDZ}$ exhibits significant promiscuity by recognizing binding partners having both *C-terminal* as well as *internal binding peptide* motifs[27,39,50]. Thus, it is expected that Dvl2$^{PDZ}$ displays divergent binding modes with its partners[35]. In the context of Wnt-PCP signalling, Dvl2 assembles with the activated Frizzled receptors and triggers RhoA by activating its GEF, i.e., WGEF. The role of Dvl2 in this signal transduction event is to activate WGEF by releasing it from its autoinhibited state. This step involves interaction between the PDZ domain of Dvl2 and WGEF. However, the particular region of WGEF that mediates its interaction with Dvl2$^{PDZ}$ was not known. Here, we have successfully identified an internal peptide motif of WGEF, lying between the N-terminal inhibitory and the DH domains, that facilitates the interaction of the protein with Dvl2 to activate the GEF. Indeed, this is one of the crucial mechanisms involved in releasing the autoinhibition of WGEF. Using MD simulations and comparative structural analysis, we show that Dvl2$^{PDZ}$ exhibits diverse modes of ligand recognition, which is perhaps true of the WGEF$^{pep}$, as well. Compared to the other known Dvl2$^{PDZ}$–*internal peptide* interactions, in the WGEF-Dvl2$^{PDZ}$ complex, only five residues at the $P_2$, $P_0$, $P_{-1}$, $P_{-2}$ and $P_{-3}$ positions appear to contribute significantly in stabilizing their interactions (Fig. 1d). Furthermore, in terms of consensus sequence, the substitution of P11A at the $P_2$ position of WGEF$^{pep}$ makes the peptide homologues to the class III PDZ binding peptide. However, in terms of the mode of binding, they differ significantly. Class III peptide bound to nNOS$^{PDZ}$ (PDB ID: 1B8Q)[51] happens to be relatively shifted towards the N-terminal of the α2 helix of the Dvl2$^{PDZ}$ domain (Supplementary Fig. 7). Therefore, just based on the sequence of binding peptide mode of binding cannot be inferred. Additionally, from MD studies we observed that WGEF$^{pep}$ and N3$^{pep}$ peptides do not seem to adopt the stringent β strand conformation, as seen in case of the crystal structures of other PDZ-*internal peptide* complexes[35]. In short, the ligand (peptide) and the Dvl2$^{PDZ}$ undergo conformational adaptation for mutual recognition. A genome-wide screen for internal PDZ binding motifs suggests that the ability of PDZ domains to bind internal peptides is much more prevalent than previously recognized[41]. To comprehend how Dvl2$^{PDZ}$ binds to the identified binding motif within the structure of hWGEF, we looked at the AlphaFold predicted structure of hWGEF (Supplementary Fig. 8a). Clearly, major portion of the structure, including the helix that connects PDZ binding motif to the NID domain, is predicted with low confidence level (Supplementary Fig. 8b). Furthermore, the spatial disposition of different domains of the structure appears to be less reliable as the loops that connect these domains are predicted with low confidence level. However, to propose a hypothesis using this model, we superimposed the part of the hWGEF structure that is predicted with higher confidence on its closest structure homologue, Leukaemia-associated RhoGEF (PDB ID: 1X86). This comparison shows that in the absence of Dvl2$^{PDZ}$ interaction, the GTPase

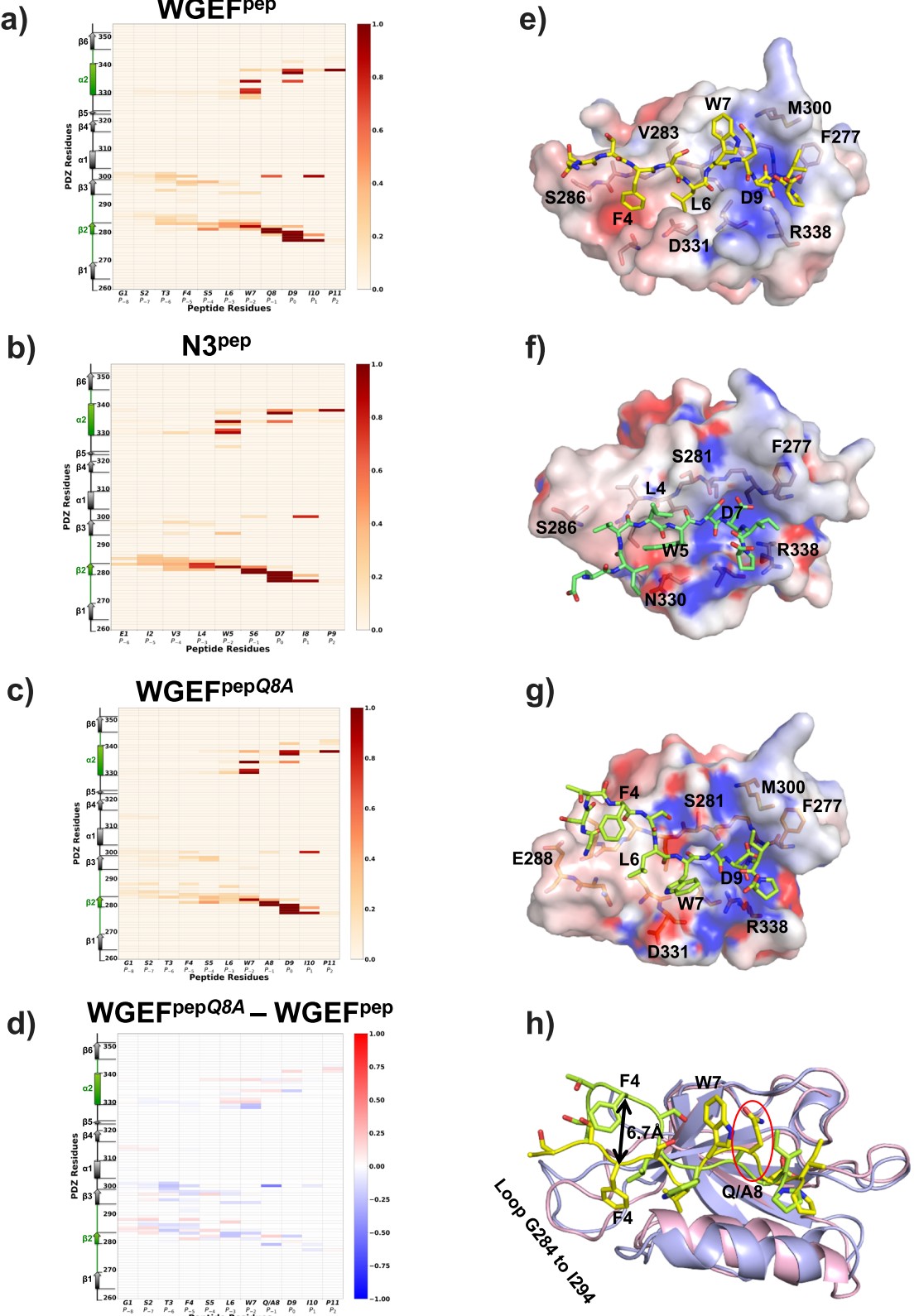

**Fig. 3 | Interaction propensity of Dvl2$^{PDZ}$ with peptide ligands. a** Interaction propensity with WGEF$^{pep}$. **b** Interaction propensity with N3$^{pep}$ and **c** interaction propensity with WGEF$^{pepQ8A}$. **d** Shows difference in the interaction propensity of WGEF$^{pepQ8A}$ and WGEF$^{pep}$. **e**–**g** show the structure of Dvl2$^{PDZ}$ (represented as surface) bound with peptide variants, WGEF$^{pep}$, N3$^{pep}$ and WGEF$^{pepQ8A}$ (represented in ball-and–sticks), belonging to one of the stable trajectories, respectively. **h** Superimposition of WGEF$^{pep}$ (yellow peptide) and WGEF$^{pepQ8A}$ (green peptide) structures.

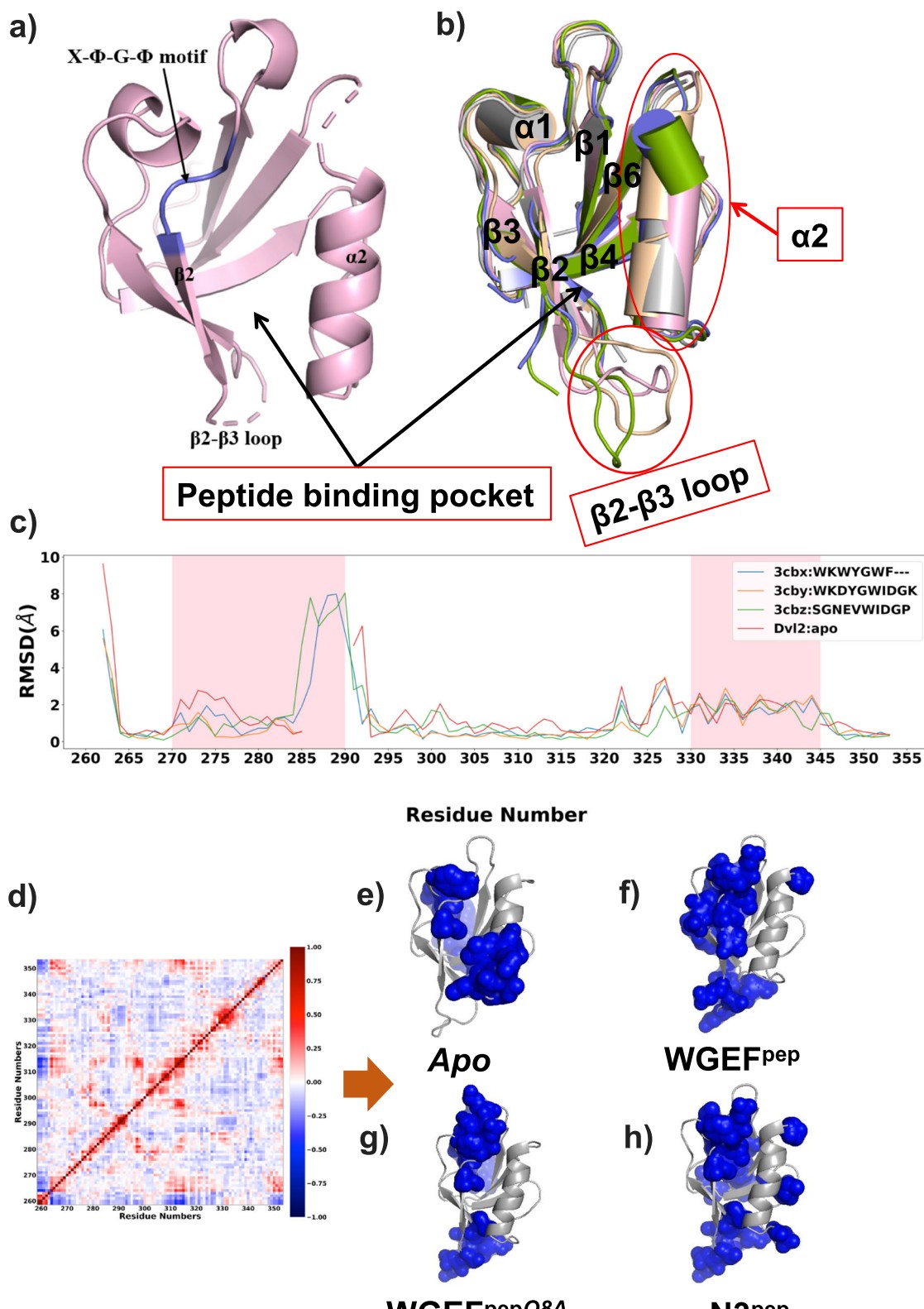

**Fig. 4 | Conformational variations between the *apo* and ligand-bound Dvl2^PDZ structures. a** *Apo* structure of human Dvl2^PDZ. **b** Superimposition of *apo* Dvl2^PDZ and peptide ligand bound Dvl2^PDZ structures colour coded as Dvl2^PDZ (PDB ID: 8WWR)-grey, C1^pep (PDB ID: 3CBX)-green, N1^pep (PDB ID: 3CBY)-blue, N2^pep (PDB ID: 3CBZ)-peach, N3^pep (PDB ID: 3CC0)-pink, major conformational deviations in the α2 helix and β2-β3 loop regions of Dvl2^PDZ are marked with a red circle. **c** Residue-

wise RMSD plot between the *apo* and *internal peptide* ligand bound Dvl2^PDZ structures. **d** Representative plot of the *dynamic cross-coupling matrix* (*cij*) used for calculating the cluster of dynamically coupled amino acids in peptide bound and *apo* Dvl2^PDZ structures. **e–h** Dynamically coupled amino acids belonging to Sector 1 were shown as blue spheres on the *apo* and WGEF^pep, WGEF^pepQ8A and N3^pep bound Dvl2^PDZ structures, respectively.

(RhoA) binding pocket of hWGEF is blocked by the N-terminal inhibitory and C-terminal SH3 domains (Supplementary Fig. 8c, d). Thus, the intra-protein interactions could render the GEF into its inactive state (Supplementary Fig. 8e). When the Dvl2$^{PDZ}$ domain engages with the "binding motif", it could introduce conformational changes in the NID, resulting in it moving away from the GTPase binding pocket of the GEF. Thus, the hWGEF–Dvl2$^{PDZ}$ association could partially release the autoinhibitory state of the former (Supplementary Fig. 8f). Perhaps, some other unknown mechanisms might also disengage the SH3 domain from the DH domain, which would result in the complete activation of GEF (Supplementary Fig. 8g). Since the putative mechanism of hWGEF activation proposed here is based on the AlphaFold predicted structure, substantive GEF activation mechanism warrants further structural studies.

In light of this, studies reported here have thus provided invaluable insights on WGEF–Dvl2$^{PDZ}$ interaction and augmented the existing repository of PDZ-*internal peptide* interaction modes. Our study may also pave the way for designing inhibitors to selectively block Wnt-frizzled signalling, which is dysregulated in many pathological conditions[52].

## Methods

### Peptides and constructs

Synthetic peptides such as WGEF$^{pep}$, WGEF$^{pep}$ mutants and N3$^{pep}$ were procured from Sai Biotech. Fused constructs of hDvl2$^{PDZ}$-WGEF$^{pep}$ were commercially synthesized from Twist Biosciences (https://www.twistbioscience.com).

### Constructs, cloning and mutagenesis

Human *Arhgef19* (encoding hWGEF 330–581) and *Xenopus Arhgef19* (encoding xWGEF 330–856) genes were PCR amplified and cloned into a modified p3E vector (with N-terminal cleavable GST tag). Human *RhoA* (encoding hRhoA 2–180) was cloned in a modified pET33b vector (with N-terminal double Strep-tag). Human Dishevelled2 PDZ (hDvl2 PDZ) was a gift from Nicola Burgess-Brown (Addgene plasmid # 38876; RRID: Addgene_38876); this vector was modified by introducing a stop codon after C354 residue of Dvl2$^{PDZ}$ to eliminate the fused peptide, cloned at the C-termini of PDZ. This construct has an N-terminal cleavable 6xHis tag. Mutations were introduced into xWGEF (330–856) through site-directed mutagenesis method and confirmed by DNA sequencing.

### Protein expression and purification

Recombinant hWGEF (330–581) and RhoA (2–180) were expressed into *Escherichia coli* (*E. coli*) strain BL21*(DE3), whereas xWGEF (330–856) was expressed into C41 (DE3) strain. Cells were grown at 37 °C in Luria Broth (LB) containing Ampicillin (100 µg/mL) followed by induction with 0.5 mM isopropyl β-D-thiogalactopyranoside (IPTG) for 16 h at 18 °C once the OD (optical density at 600 nm) of 0.6 is reached. After incubation, cells were pelleted at 4000 rpm for 20 min and stored at −80 °C. GST-tagged proteins were purified using glutathione (GSH) Sepharose-4B affinity purification followed by dialysis and GST tag cleavage using PreScission protease at 4 °C overnight. The GST tag and PreScission protease were removed by desalting, followed by a second GST affinity purification. To obtain stable and pure protein, gel filtration chromatography was performed using Hiprep 26/60 Sephacryl S-300 HR SEC column in a buffer containing 10 mM Tris–HCl pH 8, 150 mM NaCl, 2 mM DTT and 5% glycerol for xWGEF. Hiprep 16/60 Sephacryl S-200 HR SEC column with 10 mM MOPS pH 6.5, 150 mM NaCl, 2 mM DTT and 10% Glycerol buffer was used for gel filtration chromatography of hWGEF. RhoA was purified using streptactin affinity purification, after which the protein was subjected to gel filtration chromatography using Hiprep 16/60 Sephacryl S-200 HR SEC column in the buffer containing 10 mM Tris–HCl pH 8, 150 mM NaCl, 2mM EDTA, 2 mM DTT and 5% glycerol. 6xHis tagged hDvl2$^{PDZ}$ and fused hDvl2$^{PDZ}$-WGEF$^{pep}$ constructs were expressed in Rosetta (DE3) cells and purified using Ni$^{2+}$-nitrilotriacetate (Ni$^{2+}$-NTA). 6xHis tag is further cleaved by using TEV protease during overnight dialysis. TEV protease and His tag are eliminated through desalting and 2nd Ni$^{2+}$-NTA purification.

Finally, pure protein is obtained after gel filtration using Hiprep 16/60 Sephacryl S-200 HR SEC column in buffer containing 10 mM Tris–HCl pH 8, 150 mM NaCl, 2 mM DTT and 5% glycerol for hDvl2$^{PDZ}$ and 10 mM Tris–HCl pH 8, 350 mM NaCl, 2 mM DTT and 5% glycerol for hDvl2$^{PDZ}$-WGEF$^{pep}$. Mutations were introduced into xWGEF (330–856) through site-directed mutagenesis; expression and purification were carried out as described above for the GST-tagged proteins. All proteins were concentrated and stored at −80 °C for further studies.

### Protein crystallization, data collection and structure determination

Crystals of Human Dvl2$^{PDZ}$ (hDvl2$^{PDZ}$) were grown with sitting drop vapour diffusion method at 293 K by mixing 1:1 ratio of hDvl2$^{PDZ}$ (10 mg/mL) with 6 M excess of WGEF$^{pep}$ and reservoir buffer containing 0.1 M tri-sodium citrate (pH 5.6), 20% (v/v) Isopropanol, 20% (w/v) PEG4000. Crystals of Human Dvl2$^{PDZ}$ fused with WGEF$^{pep}$ (hDvl2$^{PDZ}$-WGEF$^{pep}$) were grown with sitting drop vapour diffusion method at 293 K by mixing 1:1 ratio of protein (10 mg/mL) and reservoir buffer containing 0.1 M Sodium acetate (pH 4.5) and 3 M Sodium chloride. The crystallization plate was set up using a Mosquito® crystallization robot (TTP Labtech, Royston UK). Crystals were frozen in cryoprotectants containing crystallization conditions supplemented with 28% glycerol. X-ray diffraction data was collected at 100 K on microfocus beamline ID23-2 of the European Synchrotron Radiation Facility (ESRF), Grenoble, France[53]. Crystals of the fused hDvl2$^{PDZ}$-WGEF$^{pep}$ protein were diffracted at Indus-2 synchrotron on BL-21 PX beamline at RRCAT, Indore. All the data sets were integrated with XDS[54] and scaled using AIMLESS[55,56], implemented on CCP4 software[57]. The structure was solved by molecular replacement with PHASER[58]. Human Dvl2-PDZ structure (PDB entry: 2REY) served as a search model. The structure was further iteratively built using COOT[59] and refined using PHENIX[60,61]. The structure validation was performed using Molprobidity[62]. Coordinates and structure factors for hDvl2$^{PDZ}$ and WGEF$^{pep}$ fused hDvl2$^{PDZ}$ are deposited in the Protein Data Bank under accession code: PDB IDs 8WWR and 8YR7, respectively.

### In vitro guanine nucleotide exchange assay

Guanine nucleotide exchange assays were carried out using RhoA loaded with fluorescently labelled Mant-GDP[63,64]. Briefly, 150 µM of RhoA was incubated with 5 Molar excess of fluorescent label, i.e., Mant-GDP in a buffer containing 10 mM Tris pH 8, 150 mM NaCl, 2 mM DTT, 5 mM EDTA, 0.5 mM MgCl$_2$ and 5% Glycerol. The reaction was incubated on ice for 30 min, followed by the addition of 10 mM MgCl$_2$ to stop the reaction. Excess Mant-GDP was removed through buffer exchange and Mant-GDP loaded RhoA was concentrated and stored at −80 °C. For assays, 2 µM of Mant-GDP loaded RhoA was incubated with 5 µM of xWGEF in the presence or absence of hDvl2$^{PDZ}$ (0 µM, 25 µM and 50 µM) for 30 min at 25 °C. To observe effect of the synthetic peptides WGEF$^{pep}$ (GSTFSLWQDIP) and WGEF$^{pepD9A}$ (GSTFSLWQAIP) on Dvl2$^{PDZ}$–WGEF interaction, these peptides were added to a final concentration of 200 µM in the respective reactions. Once the fluorescence was stable, excess GTP (10x of labelled GTPase) was added to start the exchange reaction. All the GEF assays were performed in a buffer containing 10 mM Tris, pH 8, 150 mM NaCl, 2 mM DTT, 10 mM MgCl$_2$ and 5% Glycerol. A decrease in fluorescence was recorded using a BioTek Cytation5 plate reader with excitation and emission of 360 and 440 nm, respectively. Fluorescence was normalized and decrease was fitted using a single exponential decay mode in GraphPad Prism. All the assays were performed in triplicates.

### Microscale thermophoresis (MST)

MST experiments were performed using Monolith NT.115 instrument from Nanotemper technologies. Dvl2$^{PDZ}$ domain was fluorescently labelled as per the instructions, using Monolith Redmaleimide 2$^{nd}$ generation-cysteine reactive label. The experimental condition consists of 10 mM Tris–HCl pH8, 150 mM NaCl, 2 mM DTT, 5% Glycerol and 0.05% Tween20. For Dvl2$^{PDZ}$-peptide affinity determination, synthetic WGEF

peptides were serially diluted (1:2) 16 times, starting from an approximate concentration of 3.4 mM. 50 nM of fluorescently labelled Dvl2$^{PDZ}$ domain was added to the serially diluted peptides, followed by incubation at 25 °C for 30 minutes. Thermophoresis signals were recorded at 20% of excitation power and 40% MST at 25 °C. Data was fitted using the $K_d$ fit model in the MO. Affinity Analysis v2.3 software[65]. Like peptides, hWGEF (330–581) protein was serially diluted to obtain a protein concentration range of 138–0.0042 μM. Experimental conditions were the same as mentioned above for peptides. All MST experiments were done in triplicates.

## Molecular dynamics (MD) simulations of hDvl2$^{PDZ}$ and interacting peptides

MD simulations were performed by using the Desmond module from the Schrödinger suite[66]. The crystal structure of *apo* human Dvl2$^{PDZ}$ (PDB code: 8WWR) was used as a template for MODELLER-based homology modelling in order to build the missing loops[67]. WGEF$^{pep}$, WGEF$^{pepQ8A}$ mutant peptide and N3$^{pep}$ peptides were modelled and docked on Dvl2$^{PDZ}$ by using the previously available internal (N3$^{pep}$) peptide structure (PDB code: 3CC0). These peptides, along with the modelled Dvl2$^{PDZ}$ structure, were further used for 1 μs MD simulation. *Apo* form of Dvl2$^{PDZ}$ was also simulated by adopting a similar methodology. All the structures were placed in a simulation box of 10 Å and further solvated using TIP3P water molecules along with Na$^+$ ions to neutralize the system[68]. Next, temperature equilibration was done at a constant temperature of 300 K using the Nosé–Hoover thermostat, and the pressure was equilibrated at 1.01 bar under NPT ensemble using Martyna-Tobias-Klein barostat[69–72]. Followed by temperature and pressure equilibration, MD runs were performed using OPLS_2005 force[73,74]. MD runs were performed in triplicates. Other parameters used for simulations are mentioned in Supplementary Table 2. RMSD plots for Dvl2$^{PDZ}$ and peptides are given in Supplementary Fig. 9.

## Dynamical coupling analysis

The dynamical cross-correlation matrix (DCCM), $dC_{ij}$[75], was calculated as

$$dC_{ij} = \langle r_i r_j \rangle - \langle r_i \rangle \langle r_j \rangle$$

where $r_i$ and $r_j$ are the position vectors of the $i^{th}$ and $j^{th}$ Cα atoms, respectively of Dvl2$^{PDZ}$, at time $t$. The R implementation of the Bio3D programme (version 2.4-1)[76] was used for the DCCM calculations. The dynamical sectors were obtained from the spectral decomposition algorithm[77] using the methodology (statistical coupling analysis) reported earlier[64,78]. Trajectories from all the triplicate MD runs were merged for dynamical coupling analysis.

## Statistics and reproducibility

Guanine nucleotide exchange assays were performed in triplicates. Fluorescence decrease was fitted using a single exponential decay mode in GraphPad Prism. All results are reported with standard deviation (±SD). Statistical significance was calculated using ordinary One-way ANOVA with multiple comparisons. * Represents adjust $P$ value with statistical significance of ****$P < 0.0001$, ***$P < 0.001$, **$P < 0.01$ and *$P \leq 0.01$. MST assays were performed in triplicates. Data was fitted using the $K_d$ fit model in the MO. Affinity Analysis v2.3 software. All results are reported with standard deviation (±SD). MD simulation runs for *apo* and peptide docked Dvl2$^{PDZ}$ were performed in triplicates. Plots reported from MD simulation studies are represented with standard deviation (±SD).

## Reporting summary

Further information on research design is available in the Nature Portfolio Reporting Summary linked to this article.

## Data availability

The atomic model of *apo* human Dvl2 PDZ is available in the Protein Data Bank (PDB) under the accession code 8WWR. The atomic model of human Dvl2 PDZ fused with WGEF$^{pep}$ (no electron density for the peptide) is available in the Protein Data Bank (PDB) under the accession code 8YR7. Initial coordinates, simulation input files and final coordinates of the MD simulation runs are available at https://zenodo.org/records/10683731[79]. The source data for graphs in the manuscript can be found in Supplementary Data 1.

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

## Acknowledgements
K.K. would like to acknowledge funding from the Department of Biotechnology, India (BT/PR12502/BRB/10/ 1387/2015). We thank Prof. Dawid Igor for generously sharing various constructs of Dvl, Frizzled and WGEF. We acknowledge the European Synchrotron Radiation Facility (ESRF), ID23-2 beamline, for the provision of synchrotron radiation facilities and we would like to thank Dr. Christoph Mueller-Dieckmann for assistance and support in using the beamline. We acknowledge BL-21 PX beamline scientists Dr. Ravindra D. Makde, Dr. Ashwani Kumar and Dr. Biplab Ghosh at RRCAT, Indore, India, for helping with data collection. The authors acknowledge the Central Instrumentation Facility (CIF), Savitribai Phule Pune University (SPPU) for the MST facility. The authors acknowledge Dr. Durba Sengupta for their inputs on MD simulations. A.O. acknowledges the Council of Scientific & Industrial Research (CSIR), India for fellowship.

## Author contributions
A.O. designed and performed all the experiments. S.M. helped with the crystallization studies. A.B. screened the initial constructs of WGEF for expression and solubility. K.K. conceptualized the problem, supervised overall work, analysed the data and wrote the paper with inputs from A.O.

## Competing interests
The authors declare no competing interests.
