## [Peer Review File · Communications Biology]

Reviewers' comments:

Reviewer #1 (Remarks to the Author):

The manuscript "Unique Internal Peptide Mediated Dishevelled 2-WGEF Interaction Activates WGEF" by Omble et al., is a well written and well performed study on the identification and characterisation of the internal PDZ binding motif in WGEF that bind to Dvl2. The study sheds light on a long standing question and does correct some of the current understanding of the interaction. As the authors did a good job with the study I only have minor comments:

Results:

Subheading A novel N-terminal conserved "internal-peptide" domain.." should better be A novel N-terminal conserved "internal-peptide" motif.." to avoid confusion between folded domains like the Dvl PDZ domain and binding motifs.

I disagree with the broad classification of PDZ specificities. The three main classes of PDZ specificities are class 1-3 (all C-terms) based on the three last aas of the motif. Residues upstream may contribute to the affinity. The internal PDZ binding motifs are typically variants of the C-terminal motifs, in the sense that they follow the main specificity determining patterns . Depending on the flexibility of the domains and the peptides internal PDZ binding motifs can be accommodated. A typical example of this is the Par6 case that the authors cited. Other more recent examples include Shank PDZ (PMID: 34235485) and Whirlin PDZ (PMID: 32971111).

The "recent" study cited by the authors (line 356) is 10 years old, so the authors might consider removing "recent" from that sentence.

Discussion

The authors should change the statement on line 337 "Therefore, unlike other PDZ domains.." as it is not true, especially given the already mentioned statement on line 356 indicating that several PDZs bind to internal motifs. More correct would be "Like some other PDZ domains.."

Line 360: The statement "Our study shall also pave ways.." should better be "Our study may also pave ways..".

Reviewer #2 (Remarks to the Author):

This manuscript aims to address the interaction between the PDZ domain of DVL2 and an internal peptide derived from WGEF. Although the binding affinity is not particularly strong, it is an interesting study. However, the presentation of the manuscript is very confusing – it is not clear how the authors

determined the complex structure. The only crystal structure associated with this manuscript is the apo PDZ domain structure, which has been known for more than 15 years. Because this issue is to be addressed, this manuscript should not move forward.

Reviewer #3 (Remarks to the Author):

The authors in the manuscript entitled “Unique Internal Peptide Mediated Dishevelled2-WGEF Interaction Activates WGEF” have reported a peptide sequence from WGEF protein (WGEFpep) which can interact with Dishevelled2 (DVL2) PDZ domain. This peptide sequence follows the rules provided in an earlier study (doi: 10.1038/nchembio.152) and binds to DVL2 PDZ domain in an internal binding mode instead of a canonical terminal binding mode. The interaction of WGEFpep -DVL2 PDZ domain is further studied using biochemical and MD simulations.

The work described in the manuscript is restricted to two in vitro assays studying the interaction between WGEFpep -DVL2 PDZ, and thus, the obtained insights are not sufficiently evolved to allow for a robust mechanistic conceptual advance in Wnt signaling or in PDZ function.

Major comments:

The AlphaFold model of human WGEF (ARHGEF19) shows that WGEFpep is a part of a loop and a helix, how would authors explain the conformational switch in WGEF protein to interact with DVL2 PDZ domain?

The authors mention that crystallisation of WGEFpep -DVL2 PDZ failed, did they consider fusing the peptide to the PDZ domain which is a common practice to solve PDZ - peptide complexes?

The authors should not use statements such as “particular interaction is sufficient to activate the GEF from its autoinhibitory state” in L173 or “Hence to obtain the structural basis of” in L188 or “These findings further validates the internal allostery of PDZ domains” in L315 because their experiments do not substantiate them.

The use of term “domain” for short peptide sequences is inappropriate.

The amino acid numbers of DVL2 PDZ should correspond to the native sequence. W117 is not part of DVL2 sequence.

<https://www.uniprot.org/uniprotkb/O14641/entry#sequences>

Minor comments:

- 1) L20-21: It should be “adapter protein Dishevelled (Dvl)”
- 2) L202: Typo Glu instead of Glutamine (Gln)
- 3) L271: Typo WGEFpepQ8A instead of WGEFpepQA8
- 4) L273: Typo confirmation instead of conformation
- 5) Manuscript needs consistent formatting of “C terminal” or “C-terminal”

Response to the reviewers' comments for Manuscript COMMSBIO-23-4220

Response to reviewers:

Reviewer #1 (Remarks to the Author):

1. The manuscript "Unique Internal Peptide Mediated Dishevelled 2-WGEF Interaction Activates WGEF" by Omble et al., is a well written and well performed study on the identification and characterisation of the internal PDZ binding motif in WGEF that bind to Dvl2. The study sheds light on a long standing question and does correct some of the current understanding of the interaction. As the authors did a good job with the study I only have minor comments:

Author response: We thank the reviewer for his/her careful evaluation of our manuscript.

2. Results:

Subheading A novel N-terminal conserved "internal-peptide" domain.." should better be A novel N-terminal conserved "internal-peptide" motif.." to avoid confusion between folded domains like the Dvl PDZ domain and binding motifs.

Author response: We thank the reviewer for pointing out this error. We have now incorporated the suggestion and accordingly revised the subheading, which now reads as:

A novel N-terminal conserved 'internal-peptide' motif of WGEF mediates its interaction with Dvl2.

3. I disagree with the broad classification of PDZ specificities. The three main classes of PDZ specificities are class 1-3 (all C-terms) based on the three last aas of the motif. Residues upstream may contribute to the affinity. The internal PDZ binding motifs are typically variants of the C-terminal motifs, in the sense that they follow the main specificity determining patterns. Depending on the flexibility of the domains and the peptides internal PDZ binding motifs can be accommodated. A typical example of this is the Par6 case that the authors cited. Other more recent examples include Shank PDZ (PMID: 34235485) and Whirlin PDZ (PMID: 32971111).

Author response: We thank the reviewer for providing constructive suggestions to improve the manuscript. We have revised the sentence and made the necessary corrections in the manuscript. Now the revised text reads as:

The peptide motifs that PDZ domains recognises are primarily *C-terminal peptide motifs*, which comprise of a stretch of 7-10 residues situated at the C-terminus of the binding partners. Apart from these, some PDZ domains can also recognise the *internal motif* that possess similar number of residues but can lie anywhere in their binding partners³¹⁻³³. Binding of PDZ with internal motif mimics that of the C-terminal peptide binding with slight variations in the residues of the peptide, involved in the binding^{33,34}.

4. The "recent" study cited by the authors (line 356) is 10 years old, so the authors might consider removing "recent" from that sentence.

Author response: We agree with the reviewer and have removed the word "recent" in the revised manuscript. In the revised manuscript it is mentioned as:

"A genome wide screen for internal PDZ binding motifs suggests that the ability of PDZ domains to bind internal peptides is much more prevalent than previously recognized⁴¹."

5. Discussion

The authors should change the statement on line 337 "Therefore, unlike other PDZ domains.." as it is not true, especially given the already mentioned statement on line 356 indicating that several PDZs bind to internal motifs. More correct would be "Like some other PDZ domains.."

Author response: As suggested by the reviewer, we have corrected the statement in the revised manuscript to "Therefore, as seen in other PDZ domains.."

6. Line 360: The statement "Our study shall also pave ways.." should better be "Our study may also pave ways..".

Author response: We thank the reviewer for this suggestion. We have now changed the sentence to "Our study may also pave ways.." in the revised manuscript.

Reviewer #2 (Remarks to the Author):

1. This manuscript aims to address the interaction between the PDZ domain of DVL2 and an internal peptide derived from WGEF. Although the binding affinity is not particularly strong, it is an interesting study. However, the presentation of the manuscript is very confusing – it is not clear how the authors determined the complex structure. The only crystal structure associated with this manuscript is the apo PDZ domain structure, which has been known for more than 15 years. Because this issue is to be addressed, this manuscript should not move forward.

Author response: We thank the reviewer for the critical evaluation of the manuscript. The authors would like to clear the confusion, that we have not determined complex structure for Dvl2^{PDZ} and WGEF^{pep}. It has been explicitly stated in the manuscript that we have not determined the structure of the complex. However, we have made extensive attempts to elucidate the structure of the Dvl2^{PDZ} - WGEF^{pep} complex, but met with limited success. Following Reviewer 3's advice, we have adopted a different approach to obtain the complex structure which included fusing of the peptide motif with the Dvl2^{PDZ} domain at its C-terminus. However, even this approach failed to yield the desired result. (Please see our response to the 3rd comment of reviewer 3).

Therefore, we used coordinates from the N3^{pep}- Dvl2^{PDZ} structure to model the WGEF^{pep} -Dvl2^{PDZ} complex and using biochemical & MD simulation studies we have validated the model thus generated (line 231- line 237). Furthermore, to highlight the divergence of WGEF^{pep} -Dvl2^{PDZ} interaction from the template, we have performed MD simulations on N3^{pep}- Dvl2^{PDZ} structure, as well. This system has also

served as a positive control in MD simulations and also for the interaction propensity calculations.

We apologise if this particular aspect was not clear in the earlier version of the manuscript. In the light of above suggestion, we have revised the main text to make it further clearer.

Regarding the *apo* PDZ domain structure: The two *apo* Dvl2^{PDZ} structures (PDB IDs: 2F0A & 3FY5) available in PDB are from *Xenopus*. However, the structure reported here is from humans. Although the structure reported here is highly homologous to *Xenopus* Dvl2^{PDZ}, the amino acid sequence of the former is slightly divergent from the former.

Reviewer #3 (Remarks to the Author):

1. The authors in the manuscript entitled “Unique Internal Peptide Mediated Dishevelled2-WGEF Interaction Activates WGEF” have reported a peptide sequence from WGEF protein (WGEFpep) which can interact with Dishevelled2 (DVL2) PDZ domain. This peptide sequence follows the rules provided in an earlier study (doi: 10.1038/nchembio.152) and binds to DVL2 PDZ domain in an internal binding mode instead of a canonical terminal binding mode. The interaction of WGEFpep -DVL2 PDZ domain is further studied using biochemical and MD simulations.

The work described in the manuscript is restricted to two in vitro assays studying the interaction between WGEFpep -DVL2 PDZ, and thus, the obtained insights are not sufficiently evolved to allow for a robust mechanistic conceptual advance in Wnt signaling or in PDZ function.

Author response: The authors express their sincere gratitude to the reviewer for his/her detailed and invaluable comments on the manuscript. We beg to differ with the reviewer’s concern regarding the in vitro assays to be ‘limited’. To validate our structural model we have performed two divergent assays that would probe the influence of WGEF- Dvl2^{PDZ} interaction on the basis of GEF activity (GEF assays) and role of individual motif residues in establishing the interaction (peptide based binding assays). We feel that these assays are adequate to address the questions being probed in the manuscript. Additionally, we have performed Molecular dynamics simulations on ‘control’ systems to further complement the experimental results. However, cell based migration assays would have augmented our findings but, we feel that, the quantum of value these assay would add to the finding is not significant. Hence, we feel that, adding other assays would be out of scope of this manuscript.

Major comments:

2. The AlphaFold model of human WGEF (ARHGEF19) shows that WGEFpep is a part of a loop and a helix, how would authors explain the conformational switch in WGEF protein to interact with DVL2 PDZ domain?

Author response: We thank the reviewer for this excellent suggestion. We have observed that the spatial dispositions of the protein domains in the predicted structure are less reliable, as the loops that connect these domains are predicted with lower confidence value. Furthermore, the conformational changes that would occur upon

binding of Dvl2^{PDZ} domain appear to be significant and can not be predicted from the AlphaFold structure of hWGEF. However, the AlphaFold structure would be good starting model to propose a hypothesis. Therefore, in the light of your suggestion we have proposed a hypothetical model that could explain the partial release of hWGEF from its auto-inhibitory state, upon binding to Dvl2^{PDZ}. We have now included this hypothesis in the “discussion” section, which would read as shown below.

To understand binding of Dvl2^{PDZ} with the identified binding motif, when the motif is part of hWGEF structure, we looked at the AlphaFold predicted structure of hWGEF (Supplementary Fig. 7a). Clearly major portion of the structure, including the helix that connects PDZ binding motif to that NID domain, is predicted with low confidence level (Supplementary Fig. 7b). Furthermore, the spatial disposition of different domains of the structure appears to be less reliable as the loops that connect these domains are predicted with low confidence level. However, to propose a hypothesis using this model, we superimposed part of the hWGEF structure that is predicted with higher confidence, on its closest structure homolog, Leukemia-associated RhoGEF (PDB ID: 1X86). This comparison shows that, in the absence of Dvl2^{PDZ} interaction, the GTPase (RhoA) binding pocket of hWGEF is blocked by the N-terminal inhibitory and C-terminal SH3 domains (Supplementary Fig. 7c and d). Thus, the intra-protein interaction of the protein could render the GEF into its inactive state (Supplementary Fig. 7e). When the Dvl2^{PDZ} domain engages with the “binding motif”, it could introduce conformational changes in the NID, resulting it to move away from the GTPase binding pocket of the GEF. Thus, hWGEF-Dvl2^{PDZ} association could partially release the autoinhibitory state of the former (Supplementary Fig. 7f). Perhaps, some other unknown mechanism might also disengage the SH3 domain from DH domain, which would result in the complete activation of GEF (Supplementary Fig. 7g). Since, the putative mechanism of hWGEF activation proposed here is based on the AlphaFold predicted structure, further substantive GEF activation mechanism warrants further structural studies.

Supplementary Fig. 7 hWGEF activation mechanism based on its AlphaFold structural model. **a)** Overall AlphaFold predicted structure of hWGEF. Regions of the structure are coloured based on the prediction confidence level. The level of increase in prediction confidence is scaled from red(low) to blue(high) **b)** The Dvl2^{PDZ} binding motif of hWGEF (shown in magenta) is zoomed **c)** Spatial disposition of DH(yellow), PH(pink), SH3(orange) and NID(brick) domains of hWGEF. The Dvl2^{PDZ} binding motif is shown in blue. **d)** Superimposition of hWGEF and Leukemia-associated RhoGEF (PDB ID: 1X86) structures. The latter is complexed with RhoA (shown in grey). **e)-g)** Putative mechanism of Dvl2^{PDZ} induced activation of hWGEF. Binding of Dvl2^{PDZ} to hWGEF could dismantle the intra-protein interaction, thereby unblocking the GTPase binding pocket of the GEF.

3. The authors mention that crystallisation of WGEF^{pep} -DVL2 PDZ failed, did they consider fusing the peptide to the PDZ domain which is a common practice to solve PDZ - peptide complexes?

Author response: We thank the reviewer for this constructive comment. Indeed, we have tried to crystallize Dvl2^{PDZ} fused with the WGEF^{pep} at its C-terminal. We

designed different Dv12^{PDZ}-linker-WGEF^{pep} constructs, expressed, purified them and subsequently tried to crystallize. The only construct that crystallized had a PDZ binding peptide (GSTFSLWQDIP) at the C-terminus of Dv12^{PDZ}, separated by a four amino acid linker (SGGG). However, this construct crystallized in a condition comprising of 3M NaCl. We screened around 3000 conditions by varying the protein concentration, pH and precipitants to obtain crystals under different conditions. However, the only condition that yielded crystals comprised of 0.1M Sodium acetate, pH 4.5 and 3M Sodium chloride. These crystals had I4₁ symmetry with one molecule present in the asymmetric-unit. The situation was similar to those observed in 3QDO (PMID: 21422294), 6SPV (PMID: 32652058), 6MTU (PMID: 31317644) and 6SPZ (PMID: 32652058). All of these peptide-fused-PDZ constructs were crystallized in I4₁ space group. However they markedly differ in their crystal packing, when compared with our structure. Another significant difference in the crystallization are their crystallization conditions, which are devoid of high salt content and most of them crystallize in presence of low molecular weight PEGs. Perhaps, these factors affect bending of the fused peptide with neighboring PDZ molecule inside the crystal system. From these comparisons we surmise that the high concentration of salt present in the crystallization condition might have impeded the protein-protein interaction (fused-peptide and PDZ interactions). These new findings have been incorporated into the revised text (Main (line numbers 191-193) as well as in the supplementary (Supplementary Fig 2 and Supplementary Table 2)) and it reads as given below:

Furthermore, we tried to obtain the WGEF^{pep} - Dv12^{PDZ} complex structure by fusing the peptide with the PDZ domain. Even this WGEF^{pep} - Dv12^{PDZ} complexation approach failed to produce the complex structure (Supplementary Fig. 2 and Supplementary Table 2).

Supplementary Fig. 2 Purification, crystallization and structure determination of WGEF^{pep} fused Dvl2^{PDZ} **a)** gel filtration profile of WGEF^{pep} fused Dvl2^{PDZ} **b)** SDS PAGE image to show the purity of peptide fused Dvl2^{PDZ} **c)** Image showing WGEF^{pep} fused Dvl2^{PDZ} crystals under UV microscope **d)** Diffraction image showing diffraction upto 3.1Å **e)** The fused Dvl2^{PDZ} crystallized in I4₁ spacegroup with ~0.6 solvent content. The crystal packing was found to be markedly different than the other reported peptide fused Dvl2^{PDZ} structures (PDB ID: 2REY, 3CC0, 3CBX, 3CBY and 3CBZ). The electron density for the fused binding motif at the C-terminal of Dvl2^{PDZ} is not visible, as this particular segment of the structure is fully exposed to the bulk solvent. Furthermore, very high concentration of the salt (3M NaCl) in the crystallization condition might have impeded the protein-protein interaction, resulting in the abrogation of interaction between the C-terminal peptide and the PDZ domain. Constructs with varying linkers and configurations and laterations in the crystallizations of the construct that yielded any crystals met with no success. The only construct that crystallized has a PDZ binding peptide (GSTFSLWQDIP) separated by four amino acid linker (SGGG).

Supplementary Table 2 Data collection and refinement statistics of *hDvl2^{PDZ}fused WGEF^{pep}* crystals:

	hDvl2^{PDZ}fused with WGEF^{pep}
Source	BL21 beamline of Indus-II synchrotron
Space Group	I 4 ₁
a (Å)	59.49
b (Å)	59.49
c (Å)	58.25
α(°)	90
β(°)	90
γ(°)	90
Resolution limits (Å)	42.07-3.19
R_{merge}	0.058(1.091)
I/s (I)	27.8(2.7)
Number of reflections	30769(5090)
Unique reflections	2071(336)
Completeness (%)	99.9(100)
Multiplicity	14.9(15.1)
CC(1/2)	1(0.731)
Resolution limits (Å)	29.75-3.00
Number of reflections	2067
Working set	1951
Test set	116
R_{work}/R_{free}	0.2770/0.3352
Number of atoms	
Protein	540
Water	-
B factors	
Protein atoms (Å²)	120.508
Water	-
RMSD from ideal values	
Bond length (Å)	0.005
Bond angles (°)	0.845
Ramachandran plot	
Preferred (%)	85.14
Allowed (%)	14.86

4. The authors should not use statements such as “particular interaction is sufficient to activate the GEF from its autoinhibitory state” in L173 or “Hence to obtain the structural basis of” in L188 or “These findings further validates the internal allostery of PDZ domains” in L315 because their experiments do not substantiate them.

Author response: We partially agree with the reviewer and we have modified the first statement in the revised manuscript, following reviewer’s suggestion. However, we feel that rest of the two statements deemed appropriate in the context of discussions being made in the manuscript. For example, in case of second statement (“Hence to obtain the structural basis of”) we have mentioned our extensive efforts to obtain the WGEF^{pep} –Dvl2^{PDZ} complex structure and when these efforts did not yield

the desired result, we modelled the complex and validated the same using biochemical assays & MD simulations. Hence the assertion made is to explain about our complimentary approaches to obtain the structural basis of WGEF^{pep} –Dvl2^{PDZ} interactions. Regarding the last comment (“These findings further validates the internal allostery of PDZ domains”), we would like to mention that ”we are trying to submit that our findings are in line with other reported studies (Stevens & He, 2022). These findings are with reference to the observations made from the MD simulations. However, in the light of these comments, we have revised these statements to bring more clarity (line numbers 138, 171-175 and 378-379).

5. The use of term “domain” for short peptide sequences is inappropriate.

Author response: We thank the reviewer for pointing out this error. We have changed the term from “domain” to “motif” for referring WGEF peptide in the revised manuscript.

6. The amino acid numbers of DVL2 PDZ should correspond to the native sequence. W117 is not part of DVL2 sequence.

<https://www.uniprot.org/uniprotkb/O14641/entry#sequences>

Author response: We thank the reviewer for pointing out this inadvertent error. We apologize for this oversight, which has now been corrected in the revised manuscript.

Minor comments:

7. L20-21: It should be “adapter protein Dishevelled (Dvl)”

Author response: We thank the reviewer for bringing this typo to our notice. We have corrected these errors in the revised manuscript

8. L202: Typo Glu instead of Glutamine (Gln)

Author response: We thank the reviewer for bringing this mistake in our notice. We apologize for this mistake. We have corrected these errors in the revised manuscript.

9. L271: Typo WGEFpepQ8A instead of WGEFpepQA8

Author response: Authors would like to thank the reviewer for pointing this. We have made the correction in the revised manuscript.

10. L273: Typo confirmation instead of conformation

Author response: We thank the reviewer for bringing this typo to our notice. We have corrected these errors in the revised manuscript.

11. Manuscript needs consistent formatting of “C terminal” or “C-terminal”

Author response: We thank the reviewer for noticing this inadvertent error. We have now consistently used ‘C-terminal’ in the revised manuscript.

REVIEWERS' COMMENTS:

Reviewer #1 (Remarks to the Author):

I thank the authors for the replies. I am satisfied with the revision.

Reviewer #3 (Remarks to the Author):

The authors addressed most of the comments raised. The main problem of this reviewer is the use of statements that are not supported by the experiments, and as mentioned by the other reviewers many spelling, grammar, punctuation mistakes. For this manuscript to be published, the authors must correct the abstract as suggested below (points 1-3), take into account and address the rest of the points, and proof-read carefully the manuscript as they should have already done.

- 1) L19: "The central event in Wnt11-Frizzled7 involved Wnt-PCP pathway" change to "The central event in Wnt-PCP pathway", since WNT11-FZD7 is not discussed further and not relevant to the present study.
- 2) L25-L27: "Furthermore, we demonstrate that WGEF-Dvl2PDZ interaction differs considerably from the reported Dvl2 PDZ-IPM interactions." No structural or other data to demonstrate such thing, change to "Furthermore, MD simulations suggest that WGEF-Dvl2PDZ interaction may be different from the reported Dvl2 PDZ-IPM interactions."
- 3) L27-L28: "Our findings have critical implications in designing drugs that can selectively block Wnt-frizzled signalling." No insights for drug design, remove the whole sentence.
- 4) L60-L62: "comprises of" is incorrect, instead write "consists of", references supporting the domain organization are missing.
- 5) L68-L69: citation needed for "In the context of Wnt-PCP pathway, the adaptor protein Dishevelled, is shown to be involved in releasing the autoinhibitory state of WGEF".
- 6) L81: change "structural" to "binding" or remove.
- 7) L84-L85; "A novel N-terminal conserved 'internal-peptide' motif of WGEF mediates its interaction with Dvl2" should be changed to "A novel N-terminal conserved 'internal-peptide' motif of WGEF mediates its interaction with Dvl2 PDZ".
- 8) L88: correct to "...at the same time to significantly enhance its GEF activity."
- 9) L95: active voice "comprise", passive voice "is comprised of" should not be used is incorrect, instead "is composed of".
- 10) L114: Typo "binding"
- 11) L121: "which comprises both" see previous comment.
- 12) L139: Rewrite as: "PDZ domains bind to C-terminal peptides, and therefore, the residue at the extreme C-terminal of the peptide..."
- 13) L141: Since previous sentence in present tense, this sentence the same, replace "were" with "are".
- 14) L156: correct "too play key roles in stabilizing" to "play a key role too, in stabilizing"
- 15) L159-L160: The sentence, "Surprisingly, xWGEF330 L407A showed increased GEF activity, which is independent of its interaction with Dvl2 PDZ (50 μ M)" is unclear. What means independent? Is it because of the low affinity interaction?
- 16) L171-L173: It's unclear, what is the meaning of "partially releasing it from the autoinhibitory state".
- 17) L173-L175: Better remove this sentence, you just repeat what you said a line before "this particular interaction promotes the activation of GEF by partially releasing it from its autoinhibitory state".
- 18) L182-L183: "Although our GEF assay based studies have shown that indeed residues at, P0 and P-3 positions of WGEFpep play crucial role in mediating WGEF-Dvl2PDZ binding" The arguments about P-

3 are contradicting and showcase how easily overstatements are made throughout the manuscript. In L160, it is stated that the P-3 mutant increased GEF activity, yet the interaction with PDZ has a very low affinity, here P-3 is supposed to be crucial for the interaction.

19) L185-L188: "Therefore, to obtain the structural basis of Dvl2PDZ-WGEF interaction, we tried to crystallize this protein complex. Despite extensive efforts, crystallization trials met with no success. Hence we resorted to crystallizing the WGEFpep in complex with Dvl2PDZ. However, this attempt did not yield the desired result, as there was no electron density for the WGEFpep in the crystal structure of WGEFpep - Dvl2PDZ complex and Dvl2PDZ was seen in its apo form." What is the difference between the bold and underlined sentences? It seems you just repeat using different words.

20) L192: "failed to produce"

21) L193-L194: No structural basis: "Therefore, we systematically designed..."

22) L214-L216: The P11A mutant peptide would represent a class III peptide, since it harbors a C-terminal carboxyl group. You should comment on that.

23) L219: "whereas leucine at P -3 position influence the peptide- Dvl2 PDZ interaction either by stabilising the complex or by enhancing the solubility of the peptide", it's unclear. L to A (P-3) mutation increases GEF activity but has a weaker affinity. How solubility of the peptide affects the interaction?

24) Same hyphenation "WGEF pep -Dvl2 PDZ" between "WGEF pep -Dvl2 PDZ".

25) L238-L255 and Fig 3a, 3b, 3c, and 3d: The description of "the residue-wise PDZ-peptide interaction plots" has peptide residues referred as P0, P2 etc whereas in the figure they are referred with the numbers.

26) L318: "comprised of" correct as suggested before.

There are also some sentences which are difficult to read and can be rephrased:

1) L112-114: "This observation is contrary to the previous studies, as the motif identified by us is situated away from the PDZ binding domains identified earlier." Please explain better what do you mean here.

2) L147: "To test whether the WGEFpep follows the "usual" internal-peptide - Dvl2 PDZ interaction, we substituted the aspartate present at the P 0 th (Fig. 1d) position of the peptide with alanine (WGEF pepD9A), as earlier studies have shown that this particular aspartate plays key role in stabilizing the interaction between Dvl2 PDZ and its effectors with internal-peptide motifs." Too long, no references.

3) L167: "Furthermore, hWGEF Y295E is shown to exist in autoinhibition free form." In an autoinhibited form, perhaps?

4) L364: "To understand binding of Dvl2 PDZ with the identified binding motif, when the motif is part of hWGEF structure, we looked at the AlphaFold predicted structure of hWGEF." Please rewrite or remove